# Automated Schedule and Cost Control Using 3D Sensing Technologies

**Ahmed R. ElQasaby ***, **Fahad K. Alqahtani *** and **Mohammed Alheyf**

Department of Civil Engineering, College of Engineering, King Saud University,
P.O. Box 800, Riyadh 11421, Saudi Arabia
* Correspondence: ahmedrashed43@yahoo.com (A.R.E.); bfahad@ksu.edu.sa (F.K.A.)

**Abstract:** Nowadays, many construction projects in KSA still struggle with cost overruns and delay in activities. Therefore, automatic monitoring approaches are needed in the construction progress monitoring domain (CPM) to address these concerns. Thus, this paper proposed a system integrating a BIM-planned model with site laser scans, as laser scanners showed massive potential in the CPM domain. The algorithms of the proposed system recognized 3D objects based on the intersection between models, alignment accuracy, and Lalonde features. The proposed system combined 3D object recognition technology with 5D information data into a 5D progress tracking system using earned value (EV) principles. The reason behind that is a lack of research regarding conducting a 5D assessment integrated BIM with 3D sensing technology in the CPM domain. The proposed system was verified using field data from a superstructure construction project where the object recognition indicators showed a 98% recall and 99% precision in recognizing 3D objects. The proposed system also used a color-coding system to address the condition of each element based on its recognition and scheduling state and address any occlusions while calculating the recognized objects. The results also revealed an automatically updated status of the project's progress in terms of schedule(4D) and cost(5D). The automated results were also validated with a manual calculation, where a slight variation (1.35%) was observed between those calculations. This system demonstrates a degree of accurate progress tracking, automatically exceeding manual performance with less computational time.

**Keywords:** automated progress tracking; 5D BIM; laser scanning; integration; EV principles

## 1. Introduction

The success of projects is evaluated through project completion within constraints of time, scope, cost, and quality. According to the KSA vision 2030 report, in 2017 alone, approximately 60% of construction projects were 20% behind schedule. In addition, more than 35% of the project time was spent collecting and analyzing data [1]. Further, approximately 15% of the construction cost was for rework activities. Therefore, time and cost were wasted in collecting and analyzing data, making as-built plans, monitoring the project, and fixing errors [1]. Therefore, researchers turned their attention to automated inspection to increase the response time of delayed activity rather than manual inspection [2,3]. Another example is that researchers were inclined to use automation methods to track and monitor the construction progress for better visualization after 2007 [4].

In this context, the construction progress monitoring domain (CPM) has developed massively in the last two decades. The exponential increase in computational capacities has allowed the architectural, engineering, and construction (AEC) industry to develop and implement automated methods in the CPM field. Lately, the development of the CPM field has depended on two primary methods: building information modeling (BIM) and 3D sensing technologies. BIM is focused on accurately establishing "as planned" 3D models. An as-planned model can also generate a spatial representation of project components.

Then, integrating a 3D model with the project's information could produce an accurate 3D BIM-based model [5,6]. The four-dimensional model was also recognized as the schedule model (4D). The scheduling model has been designated to establish the activities' sequence over time. Cost information on the project's activities is another dimension of BIM known as 5D BIM. Activities' completion and cost over time have been simulated in a virtual environment. There have been limitations to the current BIM-based cost model, such as the cash flow analysis [7]. Some researchers also identified the sixth dimension as the facility phase. However, other studies referred to the sixth dimension as sustainability and its implementation in smart cities [8].

## 2. Previous Studies

3D sensing technologies are another crucial aspect that improved immensely track and monitor the progress of construction components. These technologies included radio frequency identification (RFID), an ultra-wideband system (UWB), a global positioning system (GPS), image processing methods, and laser scanners (LS) [9]. Previously, researchers managed to assemble "as-built" models [10], where a developed as-built model was created to restore, record, and improve historic buildings. Another study investigated integrating BIM and remote sensing instruments where a BIM-based model with a laser scanner was integrated for quality control in real-time to reduce schedule and cost overrun [11].

Among the 3D sensing technologies, A laser scanner, also known as Light Detection and Ranging (LiDAR), is one of the AEC industry's most recognized technology. Laser scanning aims to map 3D objects into point cloud datasets [12]. Similarly, researchers monitored and controlled the infrastructure components using as-built data using a laser scanner [13]. Laser-based methods have also been used in recognizing construction applications such as workspace modeling, asset management, and worker tracking [14]. Another laser scanner application tracks buildings' temporary or secondary components [15]. Although there are other 3D sensing technologies, laser scanners (LS) are one of the best-fitted technologies to track and monitor the 3D status of projects accurately [16–19]. In addition, researchers used automated methods as they have lesser limitations and could save much work and time in assessing the progress of construction projects [16].

3D spatial technologies were used to monitor and control the progress in the CPM domain. Some researchers applied the RFID system to form an as-built model, while others used UWB systems [20,21]. The image processing technology was also used in the CPM field using digital images or UAVs of construction activities [22,23]. Point cloud data sets were similarly used to evaluate the progress in construction buildings through laser scanning technology [19,24]. However, researchers used more than one sensing technology for more robustness and better results; for example, a study conducted by [25] used more than one 3D sensing technology (UWB system and laser scanner). Another study used a combination of RFID and laser scanning technology [26]. Other studies have used a fusion of image processing methods and laser scanners [24,27–29].

Furthermore, some latest review articles discussed different insights to recognize knowledge gaps and recommend future directions in the CPM field. For example, the methodology in [4] applied scientometric analysis to point out a broad picture of CPM. Another example is a systematic literature survey conducted to automate indoor progress monitoring [30]. The 3D model reconstruction and geometry quality inspection were also discussed comprehensively using the point cloud datasets [31]. Meta-analysis was estimated to review the quality of studies of object recognition performance indicators in the CPM field between 2007 and 2021 [32]. Previous studies recommended the usage of 5D assessment for future research to address the gap in the BIM integrated with 3D sensing technologies in CPM applications.

Therefore, the contribution of this paper is to propose an automated construction progress tracking system for schedule and cost control. The reason behind that is the lack of research on conducting 5D assessment in the CPM domain using EV principles [4,19,30–32]. The proposed system automatically implements a 5D assessment: progress feedback regarding schedule

and cost per scan. The 5D assessment enables reviewing the progress and states the project condition through EV principles (schedule performance index, cost performance index). The proposed system outcomes were also compared to the manual system to validate the accuracy of the proposed system.

## 3. Methodology

This paper illustrates an automated progress tracking system in construction progress monitoring to assess the updated information in schedule and cost. A BIM-planned model was established from 2D shop drawings. Then, the as-built model was established by collecting scans using laser scanning technology, processing them, and registering them to a common coordinate system. The as-built model would also be evaluated and assessed to determine the quality of point cloud sets based on three KNN searching algorithms: a fixed number of nearest neighbors, a fixed neighborhood radius, and an adaptive neighborhood radius. Once the integration between those models was automatically established, two algorithms were designed and developed to recognize the as-built objects. Another two algorithms were developed to review the progress in terms of schedule and cost (4D, 5D) using EV principles. The flowchart of the proposed approach is shown in Figure 1. Further explanation of the methodological steps is provided in the following sections.

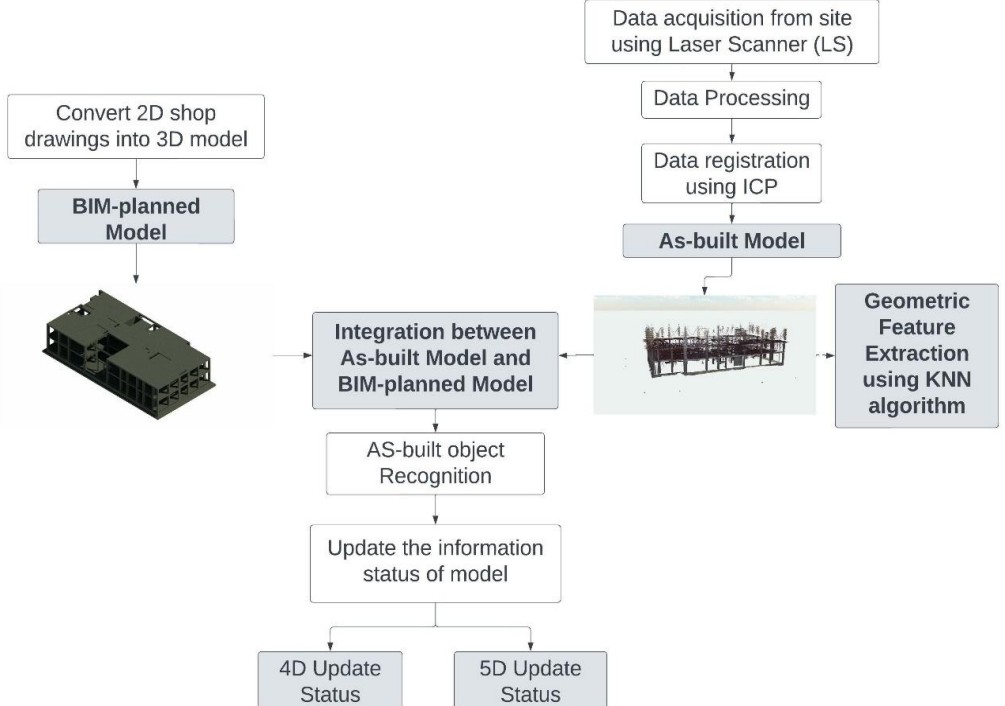

**Figure 1.** Steps of Methodology.

### 3.1. Tools

In order to apply the methodology mentioned above, a set-up of a BIM-planned model and an as-built model was crucial to be established. Revit interface was used to establish the BIM-planned model by converting 2D shop drawings to 3D models. Then, the planned schedule and cost models were manually established based on material, equipment, and labor costs for each project milestone. Material costs include supplies or materials purchased for the project, such as concrete, walls, and rebars. The transportation and storage cost is also included in the cost of materials.

The data acquisition was conducted on-site using a laser scanner Faro Focus[3D] because laser scanners are mainly accurate and efficient [33,34] (See Section 3.3). Then, datasets were processed, the scattered points were transformed into a range image, and laser scans

were registered using project reference points from one of the local coordinate systems of multiple scans to a common coordinate system [35,36]. Iteratively closest point (ICP) was then used in registration where correspondences between points of a scan called the source and points of another called the target was established to minimize the spatial distance between points in each pair. The reason behind using ICP was to achieve satisfactory registration results [37]. The outcome of the previous procedures was to generate the as-built model.

Once the as-built model was established, the next step was to indicate the strength or the weakness of the spatial distribution of the datasets by feeding the as-built model with K-nearest neighbors search algorithms (fixed number of nearest neighbors (Method I), fixed neighborhood radius (Method II), and adaptive neighborhood radius (Method III)) [38,39]. One million points were used as a reasonable sample because using the original point clouds is not computationally feasible. Firstly, the KNN algorithm based on a fixed number of neighboring points was set as [500, 5000] with an interval of 50. Secondly, the KNN algorithm based on a fixed neighborhood radius was explored using a neighborhood threshold between 5 cm–50 cm with a step length of 5 cm. The neighborhood threshold considered the further analysis of geometric features of columns, beams, and slabs when setting the threshold radius. Finally, the KNN method based on adaptive radius was set between 1–30 cm with an interval of 1cm by calculating the information entropy of the neighboring point cloud set [40]. The chosen lower band was set based on the point cloud noise, density, sensor specification, and computational constraints. However, the chosen upper band was set based on the most significant object in the scene (facades for LS-data sets) [41]. The geometric features shown in Table 1 were obtained to illustrate the spatial distribution of the datasets based on the KNN searching algorithms [41–43].

**Table 1.** Definitions of geometric features.

| Geometric Feature | Equation | Definition |
|---|---|---|
| Linear Index $L_\lambda$ | $\frac{\lambda_1 - \lambda_2}{\lambda_1}$ | represents the linear features of the neighboring point cloud clusters |
| Planar Index $P_\lambda$ | $\frac{\lambda_2 - \lambda_3}{\lambda_1}$ | represents the planar features of the neighboring point cloud clusters |
| Scatter Index $S_\lambda$ | $\frac{\lambda_3}{\lambda_1}$ | represents the scattering features of the neighboring point cloud clusters |

*3.2. Methods*

3.2.1. Three-Dimensional Object Recognition

As soon as the point cloud assessment was completed, the as-built model was incorporated into the Revit interface. Therefore, a transformation matrix was established where a planned model was fixed. The point cloud model was then transformed to match the reference model automatically. It was stated that the point cloud was clumsy enough to be recognized. Thus, the point cloud set was transformed into a geometry-based model, as mentioned thoroughly in Algorithm 1.

After the geometry-based model was established, the proposed approach was introduced to initiate this recognition system representing the correspondence between the BIM-planned model and the as-built model. The proposed approach depended on three main aspects. Firstly, the alignment accuracy between the two models was vital to object recognition. Secondly, the recognition approach was based on three distinctive features called the Lalonde features [44]. It was also used for the linerarness, surfaceness, and scatterness of a 3D point cloud set [45,46]. Finally, at least 95% of an element would intersect with the geometry model to be considered a recognized element, as declared thoroughly in Algorithm 2. In other words, Algorithm 2 searches through the BIM-planned model to find the closest geometry to each BIM-placed object. If the BIM-planned object is found, the actual component is classified based on the object type in the BIM-planned model.

---

**Algorithm 1:** Transformation of point cloud model into geometry-based model

---

*Input: Point cloud P, Point cloud model where $P \in P_M$,*
*Output: Object **O**, Geometry-based model $G_M$, Structural elements **E**,*

1   *Get Pointcloud Instance From **P** file*
2   *If pointCloudInstance ≠ null*
3   *Then*
4   *P = pointCloudInstance.GetPoint()*
5   *For Each **P** in $P_M$*
6       *O = CreateSphereSolid(**P**)*
7       *ObjectsList = append(**O**)*
8       *End $G_M$ = DirectShape.CreateElement (**ObjectList**)*
9   *End If*
10  *End*

---

**Algorithm 2**: Comparison between the geometry-based model and BIM-planned elements

---

*Input: Geometry-based model $G_M$, BIM-planned model $BIM_M$*
*Output: Structural elements **E**, Linkstructural elements $L_E$*

1   *Get **E** from **P** file*
2   *Get $L_E$ from **P** file*
3   *For Each **E** in $BIM_M$*
4       *If **E** does not intersect with $G_M$*
5       *Set Elementcolor **RED***
6       *RedList = append (E)*
7       *End If*
8   *End*
9   *For Each $L_E$ in $BIM_M$*
10      *If **E** does not intersect with $L_E$ and $L_E$ intersects with $G_M$*
11      *Set Elementcolor **Green***
12      *GreenList = append ($L_E$)*
13      *End If*
14  *End*
15      *If **E** does not intersect with $L_E$ and $L_E$ does not intersect with $G_M$*
16      *Set Elementcolor **Yellow***
17      *YellowList = append (LE)*
18      *End If*
19  *End*
20      *If YellowList >> GreenList **Then***
21      *Set Elementcolor **Blue***
22      *Bluelist = append ($L_E$)*
23      *End If*
24  *End*

---

Then, a color-coding system was established to demonstrate the condition of each element based on its recognition and scheduling state, as illustrated in Table 2. Each color would represent the recognition and scheduling state and determine whether it would be included in the calculation for automated schedule and cost.

**Table 2.** Color Coding of elements according to Algorithm 2.

|  | Recognized | Not Recognized |
|---|---|---|
| **Constructed** | the color of the material | Red |
| **Not yet Constructed** | Green | Yellow |
| **Not yet fully constructed** | Blue | Brown |

### 3.2.2. Automated Schedule and Cost Control

To update the project's status in terms of schedule and cost (4D, 5D), the authors developed two algorithms based on the results of the object recognition system. On one hand, Algorithm 3 calculated the 4D updated status based on the BCWS and BCWP estimated from the BIM-planned model and the geometry-based model, respectively, where budgeted unit cost was inserted into the algorithm. The element's color would also determine whether its cost would be included. Then, the schedule performance index (SPI) was calculated automatically to review the schedule status of a project.

---

**Algorithm 3:** Calculate the automated schedule progress

---

*Input: Structural elements $E$, Linkstructural elements $L_E$, Budget unit cost $B_{Cost}$ Concrete Volume $V_c$*
*Output: Geometry Model Cost $G_M$ Cost, BIM-planned model total cost $BIM_M$ TC, SPI*

1   *For Each Category in $E$*
2      *Get $V_c$ For category*
3      *Calculate Category cost From $B_{Cost}$ and $V_c$*
4      *$BIM_M$ TC = Category Cost*
5   *End*
6   *For Each L in Redelement*
7      *Calculate Red TC From $B_{Cost}$ and $V_{c\ Red}$*
8   *End*
9   *For Each $L_E$ in Greenelement*
10     *Calculate Green TC From $B_{Cost}$ and $V_{c\ Green}$*
11  *End*
12  *For Each $L_E$ in Yellowelement*
13     *Calculate Yellow TC From $B_{Cost}$ and $V_{c\ Yellow}$*
14  *End*
15  *$G_M$ Cost = $BIM_M$ TC — Red TC + Green TC + Yellow TC*
16  *SPI = $G_M$ Cost/$BIM_M$ TC*
17  *If SPI > 1*
18     *Then Print "Ahead of schedule."*
19  *Else If SPI < 1*
20     *Print "Behind schedule."*
21  *Else Print "Within schedule."*
22  *End If*
23  *End*

---

On the other hand, Algorithm 4 calculated the 5D updated status based on the BCWP and ACWP estimated from the geometry-based model and the revised BIM-planned model, respectively, where the actual unit cost was inserted into the algorithm. The element's color would determine whether its cost would be included. Then, the cost performance index (CPI) was calculated automatically to review the cost status of the project.

---

**Algorithm 4:** Calculate the automated Cost progress

---

*Input: Structural elements $E$, Actual unit cost $A_{Cost}$, Geometry Model Cost $G_M$ Cost*
*Output: Actual Total cost Actual TC, CPI*

1   *For Each Category in $E$*
2      *Get $V_C$ For Category*
3      *Calculate Category cost From $A_{Cost}$ and $V_C$*
4      *Actual TC = Category Cost*
5   *CPI = $G_M$ Cost/Actual TC*
6   *If CPI > 1*
7      *Then Print "Under Budget"*

---

| **Algorithm 4**: *Cont.* |
|---|
| **8**    ***Else If*** *CPI < 1* |
| **9**       │  ***Then Print*** *"Over Budget"* |
| **10**  ***Else Print*** *"On Budget"* |
| **11**  ***End If*** |
| **12**  ***End*** |

### *3.3. Case Study*

The data comprises a set of four field laser scans obtained from an investment building in The Rawda Administration Center, mainly consisting of reinforced concrete frame structure and Hardy slabs. The project location is [24.795813, 46.839646] beside Shaikh Isa Bin Salman Al Khalifah Rd, Al Maizilah, Riyadh. The site image of the case study is shown in Figure 2.

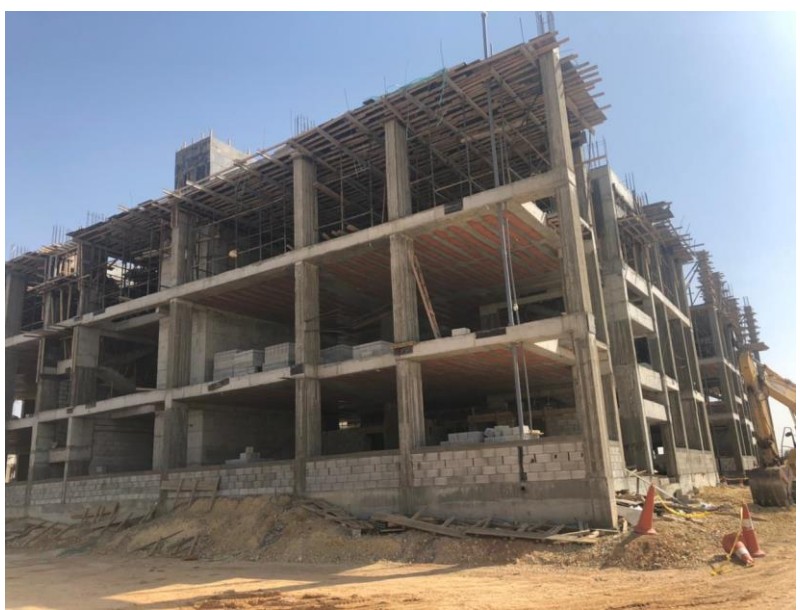

**Figure 2.** Site Image of the case study.

The construction site was scanned using Faro Focus[3D] [47] between 25 December 2020 and 20 January 2021. The weather on the days of the survey was hot; however, with a clear sky and low wind.

## 4. Results and Discussions

### *4.1. Point Cloud Characteristics*

Regarding the evaluation of registration quality, the point cloud characteristics should be thoroughly discussed. Table 3 summarizes the dataset's characteristics in all scans. The findings showed a relatively high average result of RMSE. The first reason behind that is Faro Focus 3D usually has a higher ranging error [+2 mm at 10 m and 25 m each at 90% and 10% reflectivity excluding the noise], according to the data from the manual of Faro Focus[3D] [47]. The second reason for a higher RMSE is that fewer tie points were not scanned when the scans were conducted initially. As a result, manual point matching was used, leading to a relatively higher registration error, as previously confirmed by [33]. However, the RMSE results in [19] showed a lower RMSE/Scan of 1.68 mm than the registration results conducted in this paper due to the usage of signalized targets on presurveyed site control points.



**Table 3.** Datasets characteristics in the case study.

| Scans | Scan Date | Stations/Scan | Point Cloud (Millions) | Standard Deviation $\sigma$ (mm) | RMSE (mm) | Min Overlap (%) | Inclinometer Mismatch (°) |
|---|---|---|---|---|---|---|---|
| Scan 1 | 25 December 2020 | 11 | 14.7 | 3.07 | 4.52 | 41.26 | 0.012 |
| Scan 2 | 6 January 2021 | 13 | 11.72 | 3.74 | 6.30 | 40.1 | 0.021 |
| Scan 3 | 14 January 2021 | 11 | 9.82 | 3.73 | 6 | 31.7 | 0.0735 |
| Scan 4 | 20 January 2021 | 11 | 11.22 | 3.71 | 5.5 | 35.47 | 0.021 |

The noise of the point cloud revealed an average of 3.6 mm, within the threshold of the range noise. The results also showed a minimum overlap of more than 30% for the four scans. The inclinometer mismatch error of all scans also indicated lower results, implying a good-quality scan registration within the sensor specification.

### 4.2. Point Cloud Assessment

KNN search algorithms were used to evaluate the quality of the point cloud datasets by extracting specific geometric features (linear index $L_\lambda$, planar index $P_\lambda$, and scattered index $S_\lambda$). Table 4 illustrates the geometric features obtained from the KNN search algorithms (see Section 3.1). The sample set was calculated respectively based on the eigenvalues. The results of a method I revealed that the salience features of the sample points were (linear, planar = 42.9%). However, the salience feature changed to (planar = 50%) and (planar = 74%) in methods II and III, respectively. The feature extraction values in Method III were more accurate than in Method I and Method II due to the use of entropy information that led to less unpredictability of data (more distribution).

**Table 4.** Geometric Features Extraction according to KNN methods.

| Methods | $L_\lambda$ | $P_\lambda$ | $S_\lambda$ |
|---|---|---|---|
| Method I (K = 5000) | 0.429 | 0.429 | 0.143 |
| Method II (r = 50 cm) | 0.333 | 0.50 | 0.168 |
| Method III | 0.20 | 0.740 | 0.060 |

Therefore, the results showed that the majority of the sample sets were classified as linear and planar with a small index of scatterness, which was reflected in the robust distribution of the datasets. However, previous studies pointed out that the sample set on forests was divergent, where the salience feature changed based on the type of objects. Some objects, such as stems, exhibited a linear index or planar index, while others, such as leaves, exhibited a scatterness index [39,43]. Similarly, in this paper, the results of the geometric features indicated the structure of the sample set where most structural elements were classified as linear or planar.

### 4.3. Three-Dimensional Object Recognition

The proposed approach's object recognition results were demonstrated using recall and precision rates for two incorporated scans. The recall and precision results achieved exceptionally satisfactory performance, 98%, and 99%, respectively, on average between scans. The minor errors result from objects with only a few points acquired in the scans or temporary objects with a few points wrongly recognized (false negative and false positive rates).

Figure 3 shows the object recognition results obtained from the scan on 25 December 2020, where the foundations were not recognized because the data acquisition date was after the backfilling activities, making the foundations invisible for the laser to recognize. As a result, the foundations were colored red. However, the second-floor columns showed visible progress in the schedule as the work performed exceeded the schedule; hence they

are colored green. Some of these columns failed to be recognized because of occlusions; hence they are colored yellow.

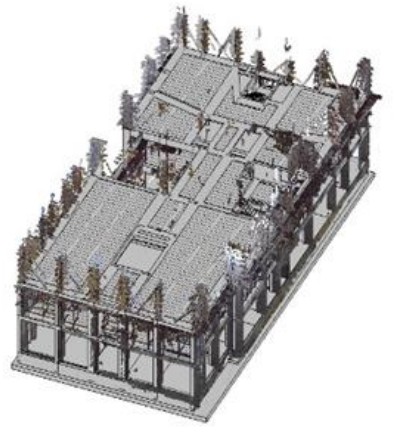 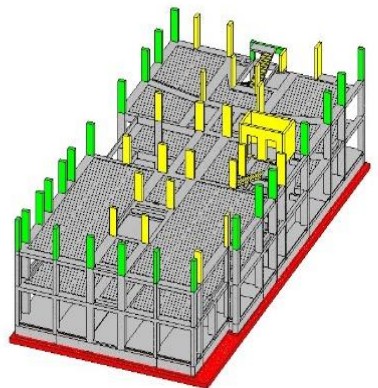

(**a**) Models before generating the code  (**b**) Models before generating the code

**Figure 3.** The object recognition results between models on 25 December 2020.

Similarly, from Figure 4, it was observed that the object recognition results obtained from the scan on 20 January 2021, where the foundations were not recognized, as mentioned in the previous paragraph. Nevertheless, the second-floor slab showed modest progress in the schedule; hence they are colored green. While the second-floor slab is not yet fully constructed; therefore, they are colored blue.

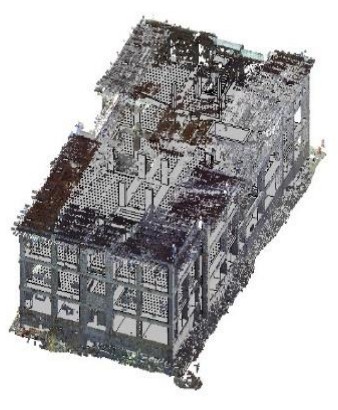 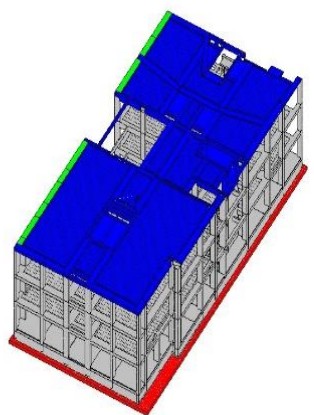

(**a**) Models before generating the code  (**b**) Models before generating the code

**Figure 4.** The object recognition results between models on 20 January 2021.

Three-dimensional object recognition is built mainly on the similarities between the attributes and properties in as-planned and as-actual models. Therefore, researchers used various approaches to recognize the 3D point clouds. Therefore, the case study findings and previous studies regarding object recognition results [17,19,37] are compared. The case study findings show a higher precision and recall than those presented by [17,37]. However, it is in agreement with the findings reported by [19].

### 4.4. Schedule and Cost Control

The proposed system generates a user interface where the calculation of the progress tracking for schedule and cost can be measured. The user interface estimates the progress of schedule and cost based on the principles of EV using the budgeted cost of work scheduled (BCWS), Budgeted cost of work performed (BCWP), and Actual cost of work performed (ACWP). Figure 5 shows the progress tracking results for the scan acquired on 25 December 2020 using the schedule and cost of concrete work in the case study. Figure 5 also shows the total cost from unrecognized elements in BCWS, BCWP, and ACWP.

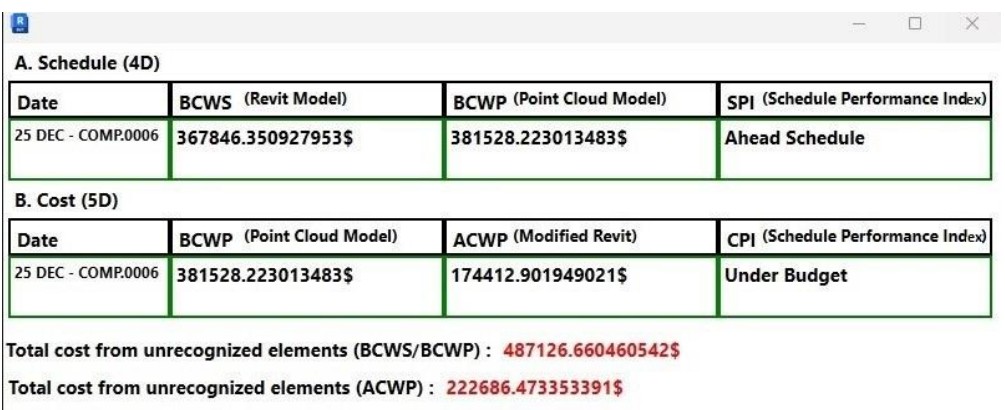

**Figure 5.** The proposed system's user interface Calculation of SPI and CPI on 25 December 2020.

However, as previously illustrated, the foundations were considered the only unrecognized elements in this case study. They must be included in the schedule and cost estimation because construction activities depend on the foundations' completion. In this case study, due to the 100% completion of the unrecognized elements (the foundations), BCWS is equivalent to BCWP as presented below Cost (5D) in Figure 5.

Table 5 demonstrates the results of the 4D progress for the case study, including the costs of unrecognized elements. Using the calculated SPI, the schedule performance of the whole project at chosen scans was determined. The results showed a fast-track project where the two scans were ahead of schedule, as the SPI was larger than one in both scans. Fewer studies were conducted to update the schedule automatically [15,27,37]. These studies mainly depended on the construction schedule to show the progress. Meanwhile, this paper demonstrates the automation of an updated schedule based on the visible recognized elements and their budget unit cost to calculate the schedule performance index.

**Table 5.** Result of the earned value (SPI) to determine the project's 4D progress (Including cost from unrecognized elements).

| Scan Date | BCWS ($) | BCWP ($) | SPI | 4D Performance |
|---|---|---|---|---|
| 25 December 2020 | 854973 | 868655 | 1.016 | Ahead |
| 20 January 2021 | 888250 | 894171 | 1.007 | Ahead |

In addition, Table 6 demonstrates the results of the 5D progress for the case study, including the costs of unrecognized elements. The cost performance of the whole project was obtained using the calculated SPI. The results showed a saving project where the two scans were under budget, as the CPI was larger than one. To the author's best knowledge, this paper is the first to address the 5D assessment in the CPM domain integrated with BIM because previous studies indicated the lack in this field, as presented in [30–32].

**Table 6.** Result of the earned value (CPI) to determine the project's 5D progress (Including cost from unrecognized elements).

| Scan Date | BCWP ($) | ACWP ($) | CPI | 5D Performance |
|---|---|---|---|---|
| 25 December 2020 | 868655 | 397099 | 2.188 | Under Budget |
| 20 January 2021 | 894171 | 408763 | 2.188 | Under Budget |

### 4.5. Comparison between Manual and Automated Calculations

To validate the accuracy of the automated user interface, the authors compare manual and automated techniques in progress calculations. Figure 6a,b show the comparison results between the manual and automated calculations on 25 December 2020 and 20 January 2021. Firstly, on 25 December 2020, the calculations are approximately similar in BCWS, but they differ in both BCWP and ACWP calculations due to the included cost of formwork and steel fixed rebar of stairs on the second floor in manual calculations. The same issue happened on 20 January 2021, where the manual result of BCWP and ACWP was slightly higher than the automated result because the manual results included the cost of the formwork of the third-floor slab.

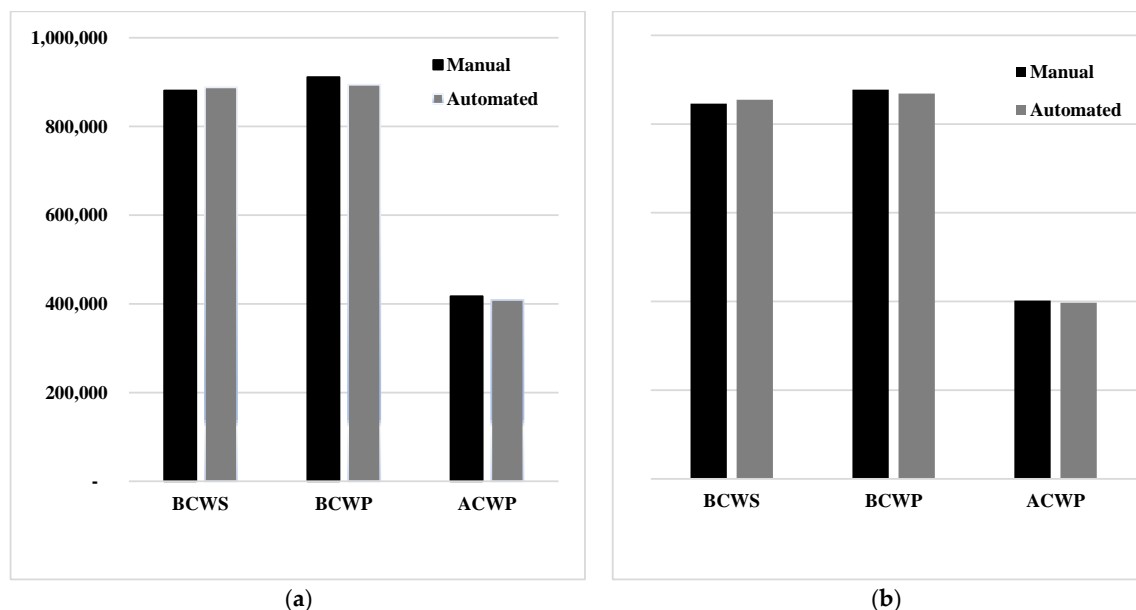

(**a**)           (**b**)

**Figure 6.** The results between manual and automated calculations on two selected dates: (**a**) 25 December 2020, (**b**) 20 January 2021.

The proposed system does not calculate the objects that are not fully constructed (colored in blue) but takes the occlusion objects that failed to be recognized (colored in yellow) into account in its estimation formula. The automated results differ only by 1.35% from the manual calculation. This variation comprises only the objects that are not fully constructed and the wrongly recognized objects.

Further, Table 7 compares manual and automated calculations regarding SPI and CPI, where the results showed a slight difference in SPI results between manual and automated calculations, even though it does not affect the project's status. At the same time, the CPI results showed approximate results between manual and automated calculations. The study findings above prove the validity of the proposed system. Hence, this system can be adapted to construction projects, enhancing the monitoring and controlling process as well as increasing the efficiency of schedule and cost updates with less time. This system can be developed outside the Autodesk platform, expanding the knowledge beyond one platform. Additionally, this system can be expanded to incorporate more domain knowledge (sus-

tainability 6D, facility management 7D). Further, this system can be conducted in different types of projects, which could highlight other factors for improvement.

**Table 7.** The results of manual and automated calculation in terms of SPI and SPI.

| Scan Date | Manual | | Automated | |
|---|---|---|---|---|
| | SPI | CPI | SPI | CPI |
| 25 December 2020 | 1.037 | 2.183 | 1.016 | 2.188 |
| 20 January 2021 | 1.034 | 2.183 | 1.007 | 2.188 |

## 5. Conclusions

This paper presented an automated progress-tracking system that integrates 5D information data with laser scanning using the data collected from a superstructure construction project. The proposed system algorithms were based on the intersection percentage between models, alignment accuracy, and Lalonde features. The proposed system also automatically estimates the construction progress and updates the project status in schedule and cost with less computational time. The main findings of the case study revealed that the object recognition indicators (recall and precision) achieved a remarkably decent performance of 98% and 99%, respectively. The proposed system also uses a color-coding system to address the different conditions of elements. Additionally, it also considers occlusions when calculating the recognized progress.

The proposed system also shows that the automated calculations of updated schedules and costs can improve progress estimation results compared to manual calculations, where there is a slight variation of only (1.35%) between manual and automated calculations. The reason is that the current system's estimation formula does not consider the cost of not fully constructed objects until they are completed. Thus, as future work, the current system should be evaluated in other construction buildings to declare a guideline and improvement. The authors acknowledge that the current approach has some limitations (i.e., the system is only available via the Autodesk Revit platform, and the laser scanner needs experienced labor). However, there is sufficient improvement using this approach to monitor the progress along with Earned Value principles.

**Author Contributions:** Conceptualization, A.R.E., F.K.A. and M.A.; methodology, A.R.E.; software, A.R.E.; validation, A.R.E., F.K.A. and M.A.; formal analysis, A.R.E., investigation, A.R.E., resources, A.R.E., F.K.A. and M.A.; data curation, A.R.E.; writing—original draft preparation, A.R.E.; writing—review and editing, A.R.E., F.K.A. and M.A.; visualization, A.R.E.; supervision, F.K.A. and M.A.; project administration, F.K.A. and M.A.; funding acquisition, F.K.A. All authors have read and agreed to the published version of the manuscript.

**Funding:** This research was funded by the Researchers Supporting Project number (RSP2023R264), King Saud University, Riyadh, Saudi Arabia.

**Institutional Review Board Statement:** Not applicable.

**Informed Consent Statement:** Not applicable.

**Data Availability Statement:** The data presented in this study are available on request from the corresponding author. The data are not publicly available due to privacy or ethical restrictions.

**Acknowledgments:** The authors extend their appreciation to the Researchers Supporting Project number (RSP2023R264), King Saud University, Riyadh, Saudi Arabia for funding this work.

**Conflicts of Interest:** The authors declare no conflict of interest.

## Abbreviations

List of abbreviations and acronyms used in the paper.

| AEC | Architectural, Engineering, and Construction |
| BIM | Building Information Modeling |
| LS | Laser scanner |
| LiDAR | Light Detection and Ranging |
| GPS | Global Positioning System |
| RFID | Radio Frequency Identification |
| UWB | Ultra-wideband |
| CPM | Construction Progress Monitoring |
| KNN | K-nearest neighborhood |
| UAV | Unmanned Aerial Vehicle |
| ICP | Iteratively Closest Point |
| PCA | Principal Component Analysis |
| RMSE | Root Mean Square Error |
| EV | Earned Values |
| SPI | Schedule Performance Index |
| CPI | Cost Performance Index |
| ACWP | Actual Cost Work Performed |
| BCWP | Budgeted cost of work Performed |
| BCWS | budgeted cost of work Scheduled |

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
