# Peer review of "Automated Schedule and Cost Control Using 3D Sensing Technologies"

_applsci, doi:10.3390/app13020783_

Round 1

Reviewer 1 Report

The work is interesting, here are some comments and requests for clarification:

-          Why not start the scanner for the Three-Dimensional Object Recognition from the beginning of the work on site and avoid “the foundations were not recognized because the data acquisition date was after the backfilling activities, making the foundation invisible for the laser to recognize”. Furthermore, in paragraph 3.4 the explanation of how the costs of the foundations have been included is not very clear.

-          Explain better: the original schedule and cost of the construction project”.

Is the original cost the estimated project budget? Surely lower than the definitive ones at the end of the construction process.

-          In the brief description of the case study “The Rawda administration Center, "mainly consisting of reinforced concrete structural elements with hollow block slab floors and rectangular columns”: Why are quotation marks used? Also, it would be better to write: …reinforced concrete frame structure and Hardy Slab.

-          The case study photo could be replaced with one that also shows the site fence, safety signs, etc.

-          There is some confusion when using the terms construction project and construction site: in the abstract is superstructure construction project while in the conclusions is superstructure construction site

-          Costs of transport of materials should also be included in the costs. (…Then, the planned schedule and cost models were manually established based on material, equipment, and labor costs for each milestone…. )

-          Finally  the abstract should be rewritten (the paper is better than how it is presented in the abstract) Some concepts/keywords are missing BIM, 4D.

Check:

a)This paper's proposed system proposed integrating a 3D-planned model with site laser scans, as laser scanners showed massive potential in the CPM domain.

b) This system demonstrates a degree of accurate progress tracking, automatically exceeding manual performance with less time and effort.  How much time you save? As for the effort, you are referring to the initial effort to set up the model: much higher than that for manual calculations but less than that required for subsequent checks and modifications.

c)…then, a color-coding system was established to demonstrate the condition of each element Perhaps it should be added:  based on its recognition and scheduling state.

Author Response

The author appreciates the reviewer's comments which enhanced the quality of the paper. So, all comments were considered in the given word file, and thus changes were marked using the track change's function. Additionally, text and formatting have been thoroughly checked to satisfy the journal requirements.

Reviewer 2 Report

Dear Authors,

Thank you for the opportunity to review your article.

The issues presented in it are currently developing quite rapidly and certainly research in this direction is essential if we want to build sustainability.

My comments relate to improving the formal structure of the article:

- it seems to me that not all items included in the bibliography - have references in the text - please verify and correct if necessary;

- I would rethink the layout, the division of the whole chapter 2. Methodology; maybe with a clear division into: materials, methods and case study. At the moment this part is not very readable for me, with this format;

- part 3 - Results and Discussion - due to the nature of the research described, I would divide this into a Results section - a clear and transparent presentation of the results of what has been explored using such tools, followed by Conclusions and Discussion - this is just my suggestion, due to the fact that the research presented is a fairly small sample. Although, and it is very good that the Authors wrote in the article, and strongly emphasised what the limitations were and that there is a need for further research.             I would also rethink the very title of the article. In my opinion, not all conclusions (4. Conclusion) follow from the study, the case study, presented in this article.

In conclusion - I would rethink the form, layout and structure of the entire article.

With best regards.

Author Response

The author appreciates the reviewer's comments which enhanced the quality of the paper. So, all comments were considered as given below, and thus changes were marked using the track change's function. Additionally, text and formatting have been thoroughly checked to satisfy the journal requirements.

Reviewer 3 Report

This paper is to propose an automated construction 100 progress tracking system for schedule and cost control. I'm really strongly interested in this article. However, the following point would contribute to improving the paper.

1- Please make there is appropriate space between the citation and the content.

2- The section of Introduction and Background make confusion to me. I suggest rewriting the section or separating the section into two sections.

3- Please cite the content of lines 35, 36, and 37 with an appropriate reference.

4- Please make sure the content for line 120 is not in italics.

5- Please make sure there are no errors in your paper. For example,  the error for lines 129 and 130.

6- The research implications are missing! I would really look at the implications to see what the papers can provide to various parties.

7- Make sure you are using appropriate references. For example, reference number 46.

Author Response

The author appreciates the reviewer's comments which enhanced the quality of the paper. So, all comments were considered in the attached word file, and thus changes were marked using the track change's function. Additionally, text and formatting have been thoroughly checked to satisfy the journal requirements.
